# Enhanced Laser-Induced Breakdown Spectroscopy for Heavy Metal Detection in Agriculture: A Review

**DOI:** 10.3390/s22155679

**Published:** 2022-07-29

**Authors:** Zihan Yang, Jie Ren, Mengyun Du, Yanru Zhao, Keqiang Yu

**Affiliations:** 1College of Mechanical and Electronic Engineering, Northwest A&F University, Xianyang 712100, China; yzh1095145815@nwafu.edu.cn (Z.Y.); jieren_rj@nwafu.edu.cn (J.R.); dumengyun@nwafu.edu.cn (M.D.); yrzhao@nwafu.edu.cn (Y.Z.); 2Key Laboratory of Agricultural Internet of Things, Ministry of Agriculture and Rural Affairs, Xianyang 712100, China; 3Shaanxi Key Laboratory of Agricultural Information Perception and Intelligent Service, Xianyang 712100, China

**Keywords:** laser-induced breakdown spectroscopy, signal enhancement, agriculture, heavy metal

## Abstract

Heavy metal pollution in agriculture is a significant problem that endangers human health. Laser-induced breakdown spectroscopy (LIBS) is an emerging technique for material and elemental analysis, especially heavy metals, based on atomic emission spectroscopy. The LIBS technique has been widely used for rapid detection of heavy metals with its advantages of convenient operation, simultaneous detection of multi-elements, wide range of elements, and no requirement for the state and quantity of samples. However, the development of LIBS is limited by its detection sensitivity and limit of detection (LOD). Therefore, in order to improve the detection sensitivity and LOD of LIBS, it is necessary to enhance the LIBS signal to achieve the purpose of detecting heavy metal elements in agriculture. This review mainly introduces the basic instruments and principles of LIBS and summarizes the methods of enhanced LIBS signal detection of heavy metal elements in agriculture over the past 10 years. The three main approaches to enhancing LIBS are sample pretreatment, adding laser pulses, and using auxiliary devices. An enhanced LIBS signal may improve the LOD of heavy metal elements in agriculture and the sensitivity and stability of the LIBS technique. The enhanced LIBS technique will have a broader prospect in agricultural heavy metal monitoring and can provide technical support for developing heavy metal detection instruments.

## 1. Introduction

Laser-induced breakdown spectroscopy (LIBS) is a material and element analysis technology based on atomic emission spectrometry [1]. LIBS produces the plasma on the sample surface by emitting a pulsed laser and then analyzes the plasma emission spectrum to determine the type and content of elements. Compared with traditional agricultural heavy metal detection methods, such as atomic fluorescence spectrometry (AFS), atomic absorption spectrometry (AAS), X-ray fluorescence spectroscopy (XRFS), and inductively coupled plasma atomic emission spectrometry (ICP-AES), LIBS has the advantages of a fast detection speed, simultaneous detection of multiple elements, and avoiding secondary contamination, which are unique advantages [2,3,4,5,6,7]. These advantages can make the LIBS technique shine in the fields of agriculture, industry, food, biology, medicine, and aerospace [8,9,10,11,12,13,14,15,16,17].

With the application and development of the LIBS technique, higher demands have been placed on its detection ability and limits of detection (LOD). In recent years, qualitative and quantitative analyses using LIBS have received wide attention from researchers. When using LIBS for qualitative and quantitative analysis of elements, it is necessary to determine the intensity of the characteristic peak of the element. The higher the characteristic peak is, the more beneficial it is to the establishment of the subsequent model [18,19]. However, due to the instability of laser energy and matrix effects of the tested samples, the obtained spectra are subject to self-absorption. The noise of the instrument and the self-absorption effect of the spectral lines would affect the intensity of the characteristic peaks, and even some characteristic peaks of trace elements would be lost [20,21]. For detecting trace elements, the stability and accuracy of the conventional LIBS technique still need to be improved [22,23]. Especially in agriculture, such as soil, water, agricultural products, fertilizer, and so on, the self-absorption phenomenon is more prominent [24,25]. Therefore, detecting trace elements in agriculture requires the LIBS signal to be enhanced. The most apparent enhancement of the LIBS signal can enhance the characteristic peak intensity [26,27] and the signal-to-noise ratio (SBR) of the spectrum [28,29]. The enhanced LIBS improves the LOD of trace elements, which provides better conditions for further analysis [30,31]. Therefore, enhancing the LIBS signal has always been a research problem.

The LIBS technique is based on the principle of using a pulsed laser to irradiate the surface of a sample to form a plasma, and then using a spectrometer to collect the characteristic spectra of the substance components emitted during the cooling of the plasma, thus enabling the detection of the sample composition and content [32]. Therefore, the key to enhancing the LIBS signal is to excite more plasma and dissipate less plasma. Only in these ways can the spectrometer pick up a stronger signal.

This review introduces LIBS’s basic instrumentation and working principle and summarizes the LIBS signal enhancement detection method of heavy metal elements in agriculture over the past 10 years. The main methods for enhancing LIBS are sample pretreatment, adding laser pulses, and using auxiliary devices. These methods significantly enhance the signal of LIBS by exciting more plasma or scattering less plasma. The enhanced LIBS signal can improve the lower limit of detection (LOD) of heavy metal elements in agriculture and improve the sensitivity and stability of the LIBS technique. The enhanced LIBS technique has a broader application prospect in agricultural heavy metal monitoring and can achieve real-time and rapid detection of heavy metal elements in agriculture. In addition, sorting out the methods to enhance LIBS signals provides a reference for improving and developing the LIBS technique.

## 2. Instrument and Principle of LIBS

The LIBS technique is an elemental analysis technology based on atomic emission spectrometry and has been rapidly developed and applied in a wide range of fields in the past few decades. The equipment of LIBS is mainly composed of a laser (or two lasers), mirror, focusing lens, motor-controlled three-dimensional sample holder, signal collector, optical fiber, spectrometer, intensified charge-coupled device (ICCD) detector, and computer [33,34].

The LIBS technique emits a high-energy laser pulse through a laser, which is reflected by a mirror and then focused on the sample surface by a focusing lens to excite the plasma. At this time, the plasma is excited, and when the laser energy is finished, these unstable plasmas will transition to low energy levels and release spectral signals of specific wavelengths. Different wavelengths in these spectral signals correspond to different elements. The generated plasma is collected by the signal receiver, converged into the optical fiber, and transmitted to the spectrometer. Then, the ICCD detector converts the optical signal into an electrical signal and transmits the electrical signal to the computer to complete the LIBS data acquisition.

After obtaining the data collected by LIBS, the data need to be analyzed. In LIBS analysis, qualitative and quantitative analysis can be performed based on the elemental spectral lines’ information and the elements’ chemical reference values. The characteristic spectral lines of specific elements can be found in the Atomic Spectral Database (ASD) of the National Institute of Standards and Technology (NIST) [35,36]. In quantitative analysis, in addition to traditional methods, such as internal and external scaling [37], machine learning methods, such as support vector machines (SVM) [38], principal component analysis (PCA) [39], the back propagation (BP) neural network [40], and partial least square (PLS) [41], are also becoming widely used. Machine learning methods, such as BP neural networks, SVM, and PCA, perform more satisfactorily than traditional methods such as PLS for LIBS spectral data mining.

In experiments using LIBS to detect heavy metal elements in agriculture, because of the instability and shortcomings of the LIBS instrument, the heavy metal elements to be measured are easily affected by the matrix effect and spectral line self-absorption, so some characteristic spectral lines are lost, and the regression model established by the above method will not be good. After using the enhancement method to enhance the LIBS signal, not only will the intensity of the characteristic spectral lines be enhanced, but the characteristic spectral lines which cannot be detected by ordinary LIBS will also be able to be detected. The quantitative effect will be significantly improved using the spectral lines obtained by the enhanced LIBS to establish the regression model.

## 3. Sample Pretreatment Enhanced LIBS for Heavy Metal Detection

Appropriate processing of the samples could improve the LIBS signal. The main methods of these treatments were changing the parameters of the sample [42], such as shape and temperature, and adding chemical reagents [43], such as nanoparticles and enrichment techniques. These methods generated more plasma in the experiment, which could effectively improve the sensitivity of LIBS. However, the experiment could only be carried out with the guidance of professionals [44]. Sample pretreatment enhanced LIBS methods for heavy metal detection in agriculture are listed in Table 1.

### 3.1. Physical Pretreatment of Samples

Changing the sample parameters, such as shape and temperature, was a typical operation of this method, and a significant feature of this method was the simplicity of the steps [45,46]. Changes in the shape and temperature of the sample could better excite the plasma.

Changing the shape of the sample allowed the laser to be better focused on the surface of the sample, optimizing the laser–sample interaction area and exciting more plasma. In order to improve the analytical sensitivity and prediction accuracy of the heavy metal element lead (Pb) in pork, Yang, et al. [47] carried out some simple treatments of drying, grinding, and tablet pressing of pork samples before using the LIBS technique to detect. After simple treatment, the contact area between the laser and the sample was increased to improve results. Therefore, some simple processing of the samples before the experiment can significantly improve the sensitivity of LIBS.

**Table 1 sensors-22-05679-t001:** Sample pretreatment enhanced LIBS for heavy metal detection.

Enhanced Methods	Sample	Elements	Evaluation Indicators	LOD	Ref.
Drying, grinding, and tablet pressing of samples	Pork	Pb	Intensity (64–2038 a.u.); R^2^ (0.200–0.960).	5.130 mg/kg	[47]
Increasing the sample temperature	Cu target	Cu	Intensity (improved 4 times); electron temperature (9000–11,500 K); electron density (0.69–0.75 × 10^17^ cm^−3^).	Improved	[48]
	Soil	Pb	Improved intensity electron temperature electron density.	3.800 mg/kg	[49]
Nanoparticle	Rice	Cd	Intensity (improved 20 times).	0.170 mg/kg	[50]
	Fruits and vegetables	Cd	Intensity (improved 2 times).	0.002 mg/kg	[51]
	Water	Cu, Pb, Cr	Cu: Intensity (improved 9 times); Pb: Intensity (improved 23 times); Cr: Intensity (improved 26 times).	Cu: 0.005 mg/L, Pb: 0.002 mg/L, Cr: 0.009 mg/L	[52]
Enrichment technology	Water	Zn	R^2^ (0.999).	4.108 mg/L	[53]
	Water	Cr	R^2^ (0.992).	0.520 mg/L	[54]
	Soil	Cr, Cr (Ⅳ)	Cr: R^2^ (0.991), RSD (7.69%); Cr (Ⅳ): R^2^ (0.993), RSD (12.98%).	Cr: 19.340 mg/kg, Cr (Ⅳ): 35.180 mg/kg	[55]
	Water	Pb, Cd, Ni	Pb: RSD (5.98%); Cd: RSD (4.25%); Ni: RSD (5.27%).	Pb: 0.001 mg/L, Cd: 0.003 mg/L, Ni: 0.002 mg/L	[56]
	Water	As, Na	As: Intensity (improved 7 times); Na: Intensity (improved 7 times).	As: 224 mg/L, Na: 18 mg/L	[57]
Repeating sample preparation	Water	Cu, Pb, Cd, Cr	Cu: Intensity (2000–4500 a.u.); Pb: Intensity (300–1400 a.u.); Cd: Intensity (500–1600 a.u.); Cr: Intensity (700–2900 a.u.).	Cu: 0.030 mg/L, Pb: 0.040 mg/L, Cd: 0.030 mg/L, Cr: 0.060 mg/L	[58]
Ultrasonic-assisted extraction technology	Rice	Pb, Cd	Pb: R^2^ (0.995), RSD (4.24%); Cd: R^2^ (0.998), RSD (2.01%).	Pb: 0.003 mg/kg, Cd: 0.044 mg/kg	[59]
Dry ashing	Leaves	Sr	R^2^ (0.990).	Improved	[60]

Note: R^2^ is determination coefficient; RSD is relative standard deviation.

In addition, increasing the sample temperature is also a way to enhance the LIBS signal. Temperature sensors were usually used in experiments to determine the raised temperature. As the sample temperature increased, the activity of molecules and atoms within the sample increased. The plasma was more easily excited when the laser focused on the sample surface at high temperatures. Thus, increasing the sample temperature improved the laser–plasma coupling, the ambient gas pressure, and the plasma plume area [61]; more plasma was produced. The way to increase the sample temperature is shown in Figure 1.

Wang, et al. [48] used the LIBS technique to ablate copper (Cu) targets at different temperatures. The electron temperature and electron density of Cu plasma increased with the increase in temperature. Umar, et al. [49] used the LIBS technique to study the relationship between sample temperature and the detection level of the heavy metal Pb in soil. With the increase in sample temperature, the emission line intensity, electron temperature, and electron density of the Pb element increased. In the obtained spectrum, there were Pb characteristic lines that were not recorded at room temperature. In addition, under the best conditions, the LOD of the element Pb reached 3.8 ppm. Therefore, the heating sample temperature could improve the LOD of trace elements in agricultural heavy metal detection by LIBS.

The above studies showed that pre-experimental sample treatment or changing sample parameters, such as shape, temperature, etc., could improve the sensitivity of the LIBS technique and the LOD of trace elements in agriculture. The trend of this method is how to quickly find the most suitable sample parameters for the experiment.

### 3.2. Nanoparticle

In addition to changing the sample’s shape and temperature parameters, adding nanoparticles to the sample could also improve the signal of LIBS. Some studies proved this point of view [62,63]. Nanoparticles had suitable dielectric and catalytic properties, which increased the activity of molecules and atoms in the sample [43]. When the laser focused on the surface of the sample coated with a layer of nanoparticles, the electric field on the surface of the sample could be increased, and the excitation threshold of the plasma was lowered, resulting in more plasma [64]. The enhancement of nanoparticles is illustrated in Figure 2. 

Due to the limited ability of the LIBS technique to detect the heavy metal element cadmium (Cd) in rice, Niu, et al. [50] put a suspension mixed with rice powder and nanoparticles on glass slides for drying and then detected and analyzed the dried samples. After treatment of the sample with nanoparticles, the spectral intensity of the element Cd was increased by 20 times; the LOD reached 0.17 μg/g. Zhao, et al. [51] used the metal nanoparticle-assisted LIBS technique to detect pesticide residues in fruits and vegetables. Compared with the common LIBS technique, the LOD of pesticide residues in fruits and vegetables using the nanoparticle-enhanced LIBS technique was two orders of magnitude lower. Yao, et al. [52] combined the LIBS technique with gold nanoparticles to detect the heavy metal elements Cu, Pb, and chromium (Cr) in liquids. The experimental results showed that the Coulomb force allowed the negatively charged AuNPs to capture heavy metal cations and directly enhanced the signal intensity. Thus, nanoparticles afforded LIBS great potential for heavy metal monitoring in agriculture.

With the aid of nanoparticles, the LIBS technique has greatly improved the sensitivity of heavy metal detection in agricultural products, such as rice, fruits, and vegetables. In addition to solids, it also has great potential for application in water quality detection. The above experiments demonstrate that nanoparticles had a significant signal enhancement effect in the detection of heavy metal elements in agriculture by the LIBS technique.

### 3.3. Enrichment Technology

The LIBS technique for detecting elements in liquids had the problem of splashing and was experimentally complex, so enrichment technology should be used to enhance the LIBS signal in liquid detection [65]. Enrichment technology could convert liquids into solid precipitates and absorb trace elements in liquid [66]. Compared with the direct detection of elements in liquid by LIBS technology, after enrichment technology, the stability of the laser-sample contact was increased, resulting in more plasma. The experimental device of enrichment technology is displayed in Figure 3.

Zhao et al. used graphite enrichment technology combined with the LIBS technique to measure and analyze the trace elements zinc (Zn) and Cr in liquid. Under the condition of using graphite enrichment technology, the spectral line emission intensity of the above two elements was obviously enhanced, and the LODs of Zn and Cr were improved, respectively [53,54]. Wu, et al. [57] used filter paper enrichment technology combined with the LIBS technique to analyze the trace element as in liquid. The stability of the LIBS signal and the LOD of the element As could be improved by enriching the filter paper and increasing the soaking time of the filter paper. Wang, et al. [56] used the electrode enrichment and LIBS technique to analyze the heavy metal elements Pb, Cd, and nickel (Ni) in water. The electrode enrichment technique could effectively improve the sensitivity of LIBS signals and the LOD of elements. Conventional LIBS techniques made it difficult to determine the different valence levels of Cr in soil, so Fu, et al. [55] used resin enrichment technology to assist the LIBS technique in detecting and analyzing Cr in soil. The experimental results showed that this method could quickly quantify the LOD of different valences of Cr.

The researchers mentioned above used enrichment technology to assist the LIBS technique in detecting heavy metals in agriculture and achieved remarkable results. The enrichment technique used graphite, filter paper, electrodes, and resin to increase the interaction between the laser and sample, which generated more plasma, enhanced the LIBS signal, and improved the stability of the LIBS technique. Enrichment technology provided a reference for detecting heavy metals in agricultural products by the LIBS technique.

### 3.4. Other Sample Processing Methods

In addition to the main methods above, some researchers have used other methods to process the samples to enhance the LIBS signal. Yang, et al. [58] realized the quantitative analysis of trace elements in an aqueous solution by repeated sample preparation. The experimental results showed that when the number of repeated samples was increased from 1 to 8, the LODs of the elements Cu, Pb, Cr, and Cd were greatly improved. Yang, et al. [59] used ultrasonic-assisted extraction technology to prepare rice samples containing heavy metals in a hydrochloric acid solution. After centrifugation, the solution was dripped on glass slides, further dried, and finally subjected to LIBS detection and analysis. Compared with the conventional method, the spectral intensity of Cd and Pb was significantly enhanced. Lazaro, et al. [60] used dry ashing to prepare samples of plant leaves and then carried out LIBS detection and analysis. The experimental results revealed that the spectral intensity of strontium (Sr) in plant leaves was significantly enhanced.

These studies demonstrate that various methods could be used to enhance the LIBS signal and provide implications for future methods of enhancing the LIBS technique.

## 4. Adding Laser Pulses Enhanced LIBS for Heavy Metal Detection

Adding laser pulses could also enhance the LIBS signal. Adding a laser pulse excited the plasma twice so that the plasma was more fully excited, and the spectrometer picked up more signals. Adding laser pulses mainly included double-pulse LIBS (DP-LIBS) [67] and LIBS assisted by laser-induced fluorescence (LIBS-LIF) [68]. Adding laser pulses enhanced LIBS methods for heavy metal detection are exhibited in Table 2.

### 4.1. DP-LIBS

DP-LIBS enhanced the LIBS signal by adjusting two laser beams. The principle of DP-LIBS was that the first laser excites the plasma, and by setting the delay time, the second laser breaks down the plasma again [69]. Compared to single-pulse laser-induced breakdown spectroscopy (SP-LIBS), DP-LIBS had two laser beams acting on the sample: one for sample ablation and the other for plasma re-excitation. The plasma was more fully excited by the action of the two laser beams. There were two kinds of double-pulses: orthogonal double-pulse LIBS (ODP-LIBS) [70] and collinear double-pulse LIBS (CDP-LIBS) [32]. The double-pulse equipment is shown in Figure 4.

**Table 2 sensors-22-05679-t002:** Adding laser pulses enhanced LIBS for heavy metal detection.

Enhanced Methods	Sample	Elements	Evaluation Indicators	LOD	Ref.
Orthogonal DP-LIBS	Soil	Mn	Intensity (improved 2 times).	Improved	[71]
	Soil	Cr	Electron temperature (improved 730 K); electron density (improved 1.8 × 10^16^ cm^−3^).	20 mg/kg	[72]
	Fertilizers	Cr	Intensity (5000–11,000 a.u.); R (0.870–0.950).	28 mg/kg	[70]
	Coptis	Cu, Pb	Cu: Intensity (5779–12,749 a.u.), R^2^ (0.974–0.993); Pb: Intensity (4703–15,838 a.u.), R^2^ (0.929–0.993).	Cu:1.910 mg/kg, Pb: 3.030 mg/kg	[73]
Collinear DP-LIBS	Sewage	Cu	Intensity (2750–4450 a.u.); R^2^ (0.993–0.999).	9.870 mg/L	[74]
	Soil	Trace elements	Intensity (improved 5 times).	Improved	[75]
	Soil	Major elements	Intensity (improved); R^2^ (improved).	Improved	[4]
LIBS-LIF	Soil	Sb	R^2^ (0.991); RMSECV (3.592 mg/kg).	0.221 mg/kg	[76]
	Rhododendron leaves	Pb	R^2^ (0.997).	1.500 mg/kg	[77]
Discharge-assisted LIBS	Water	Cr, Cu, Pb	Cr: Intensity (improved); Cu: Intensity (improved); Pb: Intensity (improved).	Cr: 1.190 mg/L, Cu: 2.640 mg/L, Pb: 3.860 mg/L	[78]

Note: RMSECV is root mean square error of cross validation.

ODP-LIBS consisted of the first laser acting on the sample surface to excite the plasma, adjusting the appropriate delay time, and the second laser perpendicular to the first laser exciting the plasma again. Du, et al. [71] analyzed the heavy metal element manganese (Mn) in soil samples using ODP-LIBS. Yu, et al. [72] used reheating ODP-LIBS to analyze ferrum (Fe), Pb, calcium (Ca), Mg, and different concentrations of Cr in soil samples. Their spectral intensity, electron density, and electron temperature increased noticeably. Compared with SP-LIBS, the LOD of Cr noticeably improved under the condition of the double-pulse. Nicolodelli et al. [70] quantified the major elements and Cr in inorganic fertilizers using a double-pulse device and compared the lower LOD with SP-LIBS. DP-LIBS significantly improved the quantification of elemental Cr. Based on the SP-LIBS experimental device, Zheng, et al. [73] built a reheating DP-LIBS experimental platform to detect the heavy metal elements Cu and Pb in Coptis. The experimental results showed that the spectral line intensities of Cu and Pb were improved under the condition of DP-LIBS. Therefore, ODP-LIBS had better detection performance.

The above was the application of ODP-LIBS in detecting heavy metals in agriculture, and the following were some applications of CDP-LIBS. CDP-LIBS consisted of the first laser acting on the surface of the sample to excite the plasma, and the second laser coinciding with the first laser exciting the plasma again by adjusting the appropriate delay time. Hu, et al. [74] used CDP-LIBS to detect the heavy metal Cu in sewage. Under the condition of the double-pulse, the spectral line emission intensity of the Cu element was noticeably increased. Nicolodelli, et al. [75] analyzed soil samples from different regions of Brazil using the CDP-LIBS technique. The combination of different wavelengths and the choice of temporal and spatial parameters were adjusted according to the experimental results to improve the LOD of trace elements by using LIBS. Furthermore, in experiments with the double-pulse, the alignment problem could be solved by using a single laser, but using multiple lasers allowed for more consideration in parameter selection. Zhang, et al. [79] quantitatively analyzed the mineral elements in roots, stems, and leaves of mature rice based on CDP-LIBS, and described the distribution of mineral elements in mature rice. He, et al. [4] used CDP-LIBS and SP-LIBS to analyze the major and micronutrient elements in soil samples, respectively. CDP-LIBS had a stronger spectral signal and better signal stability than SP-LIBS.

The above was the application of double-pulse technology to enhance spectral signals in the detection of heavy metals in agriculture. However, they only compared ODP-LIBS and CDP-LIBS with SP-LIBS, lacking a comparison between ODP-LIBS and CDP-LIBS. Comparing ODP-LIBS with CDP-LIBS made the double-pulse-enhanced detection of heavy metals in agricultural products more comprehensive and convincing.

### 4.2. LIBS-LIF

LIBS-LIF emitted a specific wavelength of laser to excite specific particles in the plasma, and the specific particles in the plasma resonated with the specific wavelength of laser to be excited again [80]. After the effect of two laser beams, the specific elements in the plasma were excited more completely. This fluorescence spectrum reduced the interference between spectral lines and had the characteristics of a high enhancement multiple and high selectivity [81]. The laser-induced fluorescence equipment is shown in Figure 5.

In order to solve the problem of the low sensitivity of LIBS in detecting trace elements due to complex spectral lines, Yi, et al. [82] used LIBS-LIF to selectively enhance the intensity of some spectral lines. After enhancing the spectral line of Pb I 405.78 nm (I was an atomic spectral line), LIBS-LIF could eliminate interference lines and improve LIBS detection. Gao, et al. [76] employed the LIBS-LIF technique to selectively enhance the characteristic lines of antimony (Sb) under the optimal parameters. LIBS-LIF significantly improved the accuracy of the quantitative model. Zhu, et al. [77] used LIBS-LIF to determine Pb in rhododendron leaves. The signal intensity of the Pb spectral line was enhanced, and the detected Pb content was between 1.5 and 2.8 mg/kg. This experiment provides a new technology for the real-time and rapid detection of Pb elements in rhododendron leaves.

Fluorescence assistance could select a specific element and enhance the LIBS signal of the element, increase the intensity of characteristic spectral lines, reduce the interaction between spectral lines, significantly improve quantitative analysis performance, and have the potential to achieve rapid and accurate analysis of heavy metals in agriculture. The above studies showed that induced fluorescence-assisted laser-induced breakdown spectroscopy was an effective method for enhancing the LIBS signal.

### 4.3. Other Methods of Optimizing LIBS Systems

In addition to the main methods above, some researchers have used other methods of optimizing LIBS systems to enhance the LIBS signal.

Changing the parameters of the LIBS instrument was also a way to enhance LIBS signals. The laser energy and the distance between the lens and the sample were critical parameters [83,84]. Chen, et al. [85] used the LIBS technique to generate laser-excited soil samples and studied the effect of the laser energy from 100 to 500 mJ on plasma. The results showed that the intensity and SBR of plasma lines were improved when the laser energy was 200 mJ. Then, they adjusted the defocused distance to observe its effect on the spectral line intensity. When the defocusing distance was +6 mm, the spectral lines of some elements were nearly doubled. This study provided theoretical support for the detection of trace elements in soil.

Discharge enhancement could also enhance the LIBS signal. The laser-induced plasma triggers the discharge of the high-voltage electrode, which in turn secondarily excites plasma [86]. Wang et al. [78] used discharge-assisted laser-induced breakdown spectroscopy to detect and analyze the trace elements Cr, Cu, and Pb in an aqueous solution. The experimental results showed that the LODs of the trace elements Cr, Cu, and Pb were improved compared with no discharge-assisted LIBS (DA-LIBS).

The above researchers used methods different from the public. These methods achieved the purpose of enhancing LIBS signals, improved the sensitivity and stability of the LIBS technique, and proved that many methods could enhance LIBS signals.

## 5. Adding Auxiliary Devices Enhanced LIBS for Heavy Metal Detection

Enhancing the LIBS signal in this way was usually achieved by adding auxiliary devices. Adding auxiliary devices where the plasma was collected reduced plasma scattering, and the spectrometer could then collect more LIBS signals. This mainly includes spatial constraint [87] and magnet device auxiliary constraint [88]. Adding auxiliary devices enhanced LIBS methods for heavy metal detection are enumerated in Table 3.

### 5.1. Spatial Constraint

The spatial constraint was a widespread common method for the physical enhancement of LIBS signals. It only required the addition of structures, such as constrained cavities or parallel plates around the sample, which had the advantage of simplicity and low cost [89]. The main principle of the spatial constraint was that the plasma produced by spatial constraint greatly increased the collision probability of the plasma, improved the space-time stability of the plasma, and reduced the scattering of the plasma, thus enhancing the LIBS signal [90]. Spatial constraint mainly included the cylinder spatial constraint [91] and hemispherical spatial constraint [83]. Some spatial constraint devices based on LIBS are shown in Figure 6.

These spatial constraint devices often need to be designed by researchers. Popov [92] made a small cylindrical cavity to analyze arsenic (As) in soil. Because the shock wave reflected by the cavity wall collides with the plasma, the spectral line intensity of As in the soil increased noticeably. Zhao, et al. [93] used femtosecond laser-induced breakdown spectroscopy to study the spatial limiting effect of Pb in soil in a cylindrical cavity. Cross-validation by LOD, relative standard deviation (RSD), and root mean square error (RMSE) of the Pb showed that the cylindrical cavity improves the sensitivity and accuracy of femtosecond laser-induced breakdown spectroscopy. Adding the constraint of a cylindrical structure enabled the better use of LIBS to detect trace elements. Meng, et al. [94] designed an easy-to-install small volume hemispherical space limiter for monitoring and analyzing heavy metals in soil. With the aid of this device, the LOD of heavy metal elements in the soil was lower.

**Table 3 sensors-22-05679-t003:** Adding auxiliary devices enhanced LIBS for heavy metal detection.

Enhanced Methods	Sample	Elements	Evaluation Indicators	LOD	Ref.
Cylinder spatial constraint	Soil	As	Intensity (improved 3–5 times).	Improved	[92]
	Soil	Pb	R^2^ (0.983); RSD (4.98%); RMSECV (0.45%).	8.850 mg/kg	[93]
Hemispherical spatial constraint	Soil	Cd, Cu, Ni, Pb, Zn	Cd: Intensity (improved 2–3 times); Cu: Intensity (improved 2–3 times); Ni: R^2^ (0.992); Pb: R^2^ (0.996); Zn: Intensity (improved 2–3 times).	Cd: 4.580 mg/kg, Cu: 3.210 mg/kg, Ni: 6.240 mg/kg, Pb: 4.540 mg/kg, Zn: 2.600 mg/kg	[94]
V-shaped spatial constraint	Soil	Cd	Intensity (206–510 a.u.); R^2^ (0.972).	0.123 mg/kg	[95]
Conical spatial constraint	Soil	Cr	Intensity (improved 0.07–0.15 times); RSD (<10%); R^2^ (0.991–0.998).	18.850 mg/kg	[96]
Magnetic field	Pb target	Pb	Intensity (improved ~2.8–~4.2 times).	Improved	[97]
	Soil	Cr	Intensity (improved 8 times).	7.700 mg/kg	[98]
	Soil	Cu, Pb	Cu: Intensity (improved ~7.7 times), electron temperature (improved), electron density (improved); Pb: Intensity (improved ~7.7 times), electron temperature (improved), electron density (improved).	Cu: 4.100 mg/kg, Pb: 1.400 mg/kg	[99]
MA-LIBS	Soil	Cd	Intensity (improved 9–27 times).	2.160 mg/kg	[100]

In addition, Fu, et al. [95] designed a V-shaped spatial constraint to detect and analyze the heavy metal element Cd in soil. The experimental results showed that this device improved the LOD of the element Cd to 0.123 mg/kg. Lin, et al. [96] employed a conical spatial constraint to analyze Cr in soil. The LOD of Cr with spatial limitation was improved. Therefore, the shape of the spatial constraint was no longer a single cylindrical and hemispherical constraint, and it could be extended to other shapes.

From the above studies, it can be seen that using spatial constraints could significantly enhance the LIBS signal. Spatial constraint methods had the advantages of a simple structure, convenient manufacture, and so on. In addition to the main cylindrical and hemispherical constraints, some new shapes, such as V-shaped spatial constraints, could also enhance LIBS signals. The innovation of spatial constraint shape according to the actual situation was the next development trend of the spatial constraint-enhanced LIBS signal.

### 5.2. Magnetic Field Assist

The magnetic field constraint was also a popular method to enhance LIBS signals [101]. The main principle of the magnetic field-constrained enhancement of the LIBS signal was to enhance the LIBS signal by increasing the excitation rate of electrons in the sample to be tested and then increasing their collision rate [102]. Therefore, magnetic field confinement played a confining role in achieving a reduction in plasma dissipation, and it increased the plasma excitation rate and the number of plasmas. The magnetic field constraint is shown in Figure 7. Akhtar, et al. [97] used laser-induced breakdown spectroscopy to study the characteristics of Pb plasma with or without an external magnetic field. The experiment proved that the external magnetic field could increase the velocity of the plasma plume and then increase the plasma’s temperature and electron number density. Therefore, the magnetic field was an effective method to enhance the detection of heavy metal signals by LIBS.

Akhtar, et al. [98] combined a magnetic field with LIBS to analyze the heavy metal elements in soil under atmospheric conditions. In the presence of a magnetic field, the enhanced signal intensity was due to the increased collision rate, the enhanced electron density was due to the magnetic field confinement, and the increased electron temperature was due to magnetic compression. In subsequent studies, Akhtar, et al. [103] used LIBS to analyze soil samples qualitatively and quantitatively. This experiment demonstrated that the external magnetic field did not affect the quantitative analysis of the sample. Then, Akhtar, et al. [99] analyzed Cu and Pb in the soil by applying a magnetic field. The experimental results showed that the LOD of Cu was improved from 12 ppm to 4.1 ppm, while the LOD of Pb was improved from 3.9 ppm to 1.4 ppm. The combination of LIBS with a magnetic field could improve the spectral intensity of trace elements in addition to the spectral intensity of the major constituents in the soil.

Using external magnetic field assistance devices by the Akhtar team could improve the sensitivity and stability of LIBS and improve the LOD of heavy metals in agriculture. Combining the external magnetic field assistance device with the LIBS technique expanded the application of the LIBS technique for heavy metal detection in agriculture.

### 5.3. Other Auxiliary Device

In addition to the above methods that could enhance the LIBS signal, some researchers have used other ways to enhance the LIBS signal.

Hu, et al. [100] used LIBS and microwave-assisted LIBS (MA-LIBS) to detect contaminated rice samples in the laboratory. Compared with the traditional LIBS method, the emission intensity of the Cd spectral line and the detection sensitivity were enhanced.

This method achieved the purpose of enhancing LIBS signals, improved the sensitivity and stability of the LIBS technique, and proved that many methods could enhance LIBS signals.

## 6. Conclusions and Future Prospect

With the advantages of simultaneous detection of multi-elements, wide range of elements, remote control, simple sample preparation, no sample pretreatment, no requirement for the state and quantity of samples, and almost no damage to samples, the LIBS technique has been widely used in agricultural detection. However, due to the complexity of elements in agricultural products, it is challenging to detect heavy metal elements by the LIBS technique. Therefore, the method of enhancing the LIBS signal has been paid more and more attention. In this study, we introduce the latest progress in detecting heavy metal elements in agriculture using enhanced LIBS signals. This includes the basic composition and principle of the instrument, the methods of sample pretreatment (physical pretreatment of samples, nanoparticle-assisted enhancement, enrichment technology, and other sample processing methods), adding laser pulses (DP-LIBS enhancement and LIBS-LIF enhancement, as well as other methods of optimizing LIBS systems), and using auxiliary devices (solid confinement enhancement, magnetic field-assisted enhancement, and other auxiliary devices) to enhance LIBS signals. Related studies have proved that using the above enhancement methods can improve the spectral line intensity, electron temperature, electron density, and SBR, thus improving the LOD of heavy metal elements in agriculture and ultimately achieving the purpose of monitoring the content of heavy metal elements in agriculture.

For the detection of heavy metal elements in agriculture, reducing the influence of other elements and obtaining a more stable plasma with higher spectral intensity are key. In particular, the wide variety of elements in agriculture is influenced by the substrate. The single enhancement method cannot enhance the LIBS signal very well, so various enhancement methods can be used together to achieve the final experimental purpose.

In LIBS experiments, determining how to excite more plasma and reduce the scattered plasma is key to enhancing the LIBS signal. The enhanced LIBS technique has a broader application prospect in heavy metal detection in agriculture. Therefore, it is important to study and develop the enhancement module with portability for LIBS instruments. Currently, most of the existing LIBS instruments are analyzed in the laboratory, which brings inconvenience to practical applications. Portable LIBS instruments can perform detection and analysis in the field, so portable LIBS instruments will be the key to subsequent development. Exploring the signal enhancement methods of portable LIBS instruments will be the ultimate trend.

## Figures and Tables

**Figure 1 sensors-22-05679-f001:**
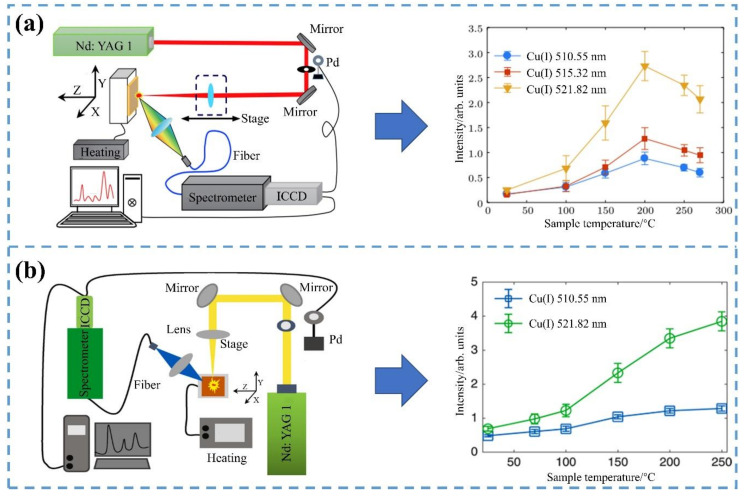
Increasing the sample temperature to enhance the LIBS signal. (**a**) Three characteristic lines of Cu element; (**b**) two characteristic lines of Cu element.

**Figure 2 sensors-22-05679-f002:**
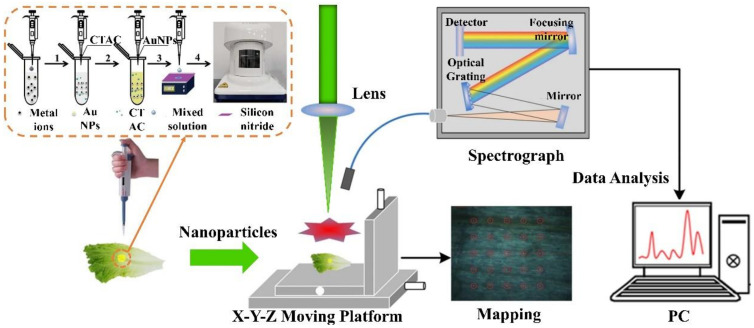
Enhancement of LIBS signal by nanoparticles. The process of sample making and experimental equipment for nano-particle enhancement.

**Figure 3 sensors-22-05679-f003:**
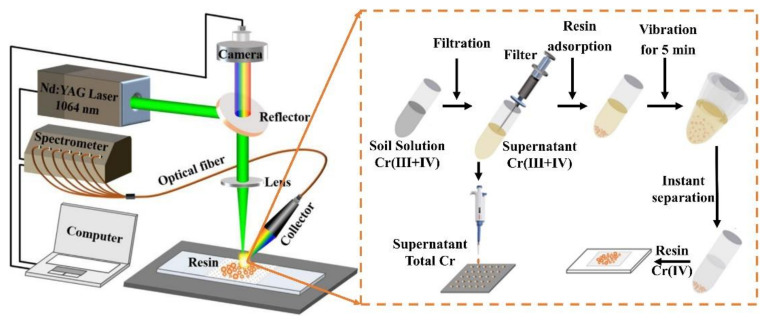
The experimental device of enrichment technology.

**Figure 4 sensors-22-05679-f004:**
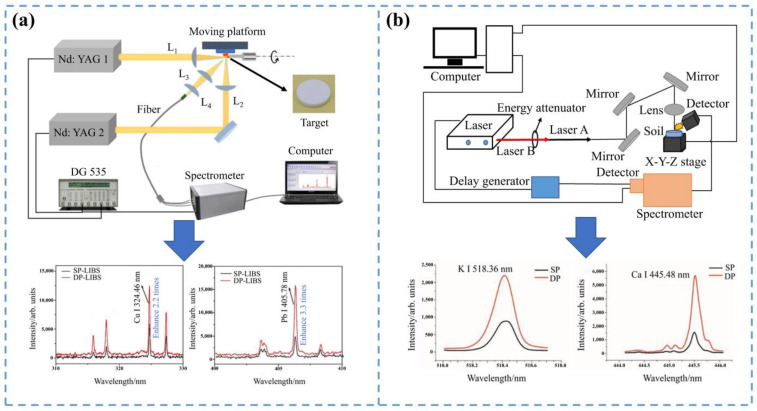
The double-pulse-enhanced LIBS signal device. (**a**) ODP-LIBS; (**b**) CDP-LIBS.

**Figure 5 sensors-22-05679-f005:**
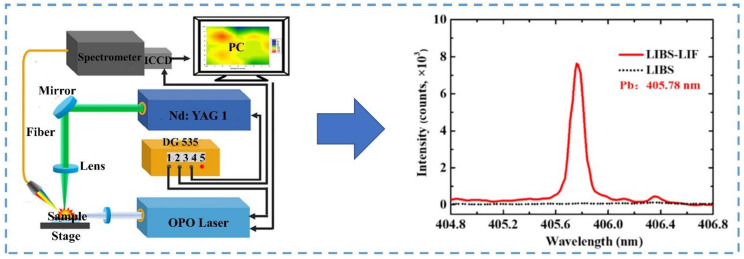
A device for combining LIBS with laser-induced fluorescence.

**Figure 6 sensors-22-05679-f006:**
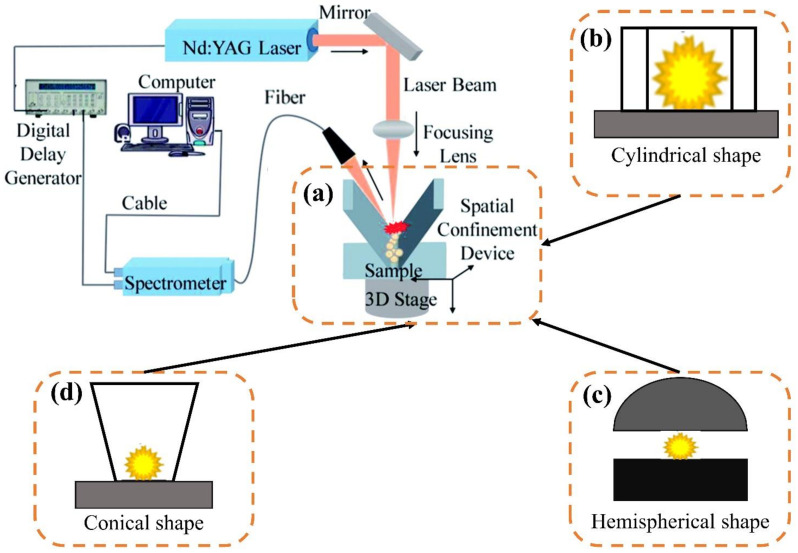
The spatial constraint-enhanced LIBS signal device. (**a**) V-shaped, (**b**) cylindrical shape, (**c**) hemispherical shape, (**d**) conical shape.

**Figure 7 sensors-22-05679-f007:**
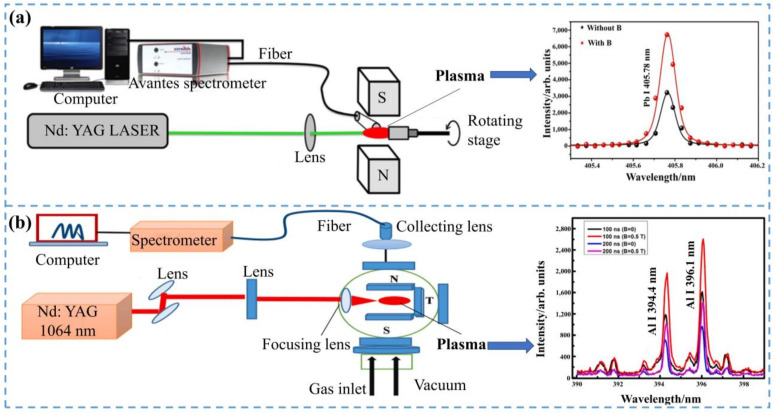
The magnetic field constraint-enhanced LIBS signal device. (**a**) Detection of Pb element; (**b**) detection of Al element.

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
