# Peer review of "Enhanced Laser-Induced Breakdown Spectroscopy for Heavy Metal Detection in Agriculture: A Review"

_sensors, 2022, doi:10.3390/s22155679_

Round 1

Reviewer 1 Report

In this work, authors illustrated the problems of LIBS technology in heavy metal detection in agriculture, introduced the methods of enhanced LIBS technology in terms of principles, and summarized the applications of the enhanced LIBS in heavy metal detection in agriculture. In my opinion, the field of writing is pertinent to this journal, the classification of enhancement methods in the review is relatively novel, the methodological discussion is detailed, and this work has wide application potential. Therefore, I think this paper is suitable for publication with some small modifications. Here are some suggestions and comments:

1. Note the details of the abbreviation, line 16 "limits of detection" should be changed to "LOD".

2. Note the correctness of the tense. In line 21, "has" should be changed to "will have".

3. In line 38, you should explain in detail whether the loss of characteristic peaks is characteristic of trace elements or whether the characteristic peaks of trace elements are more affected.

4. The "and" in line 108 should be deleted.

5. In line 121 you need to explain in detail what kind of sample parameters to find.

6. In line 149, the stability of the laser-sample contact is increased rather than the area.

7. In line 208, it should be further clarified that orthogonal double-pulse have better detection performance than what.

8. All tables in the review show the enhanced LIBS method and what the results are, but are not specific enough, so hopefully it can be added whether portable measurements are possible.

9. Note the uniformity of wording in the review. In line 224, "-" should be added between "single pulse" and in line 226, "-" should be added between "double pulse".

10. While the paper is well structured, there are several sentences difficult to understand because of improperly used English. The authors need to have this paper edited by someone proficient in technical English.

In all, the work is well done, and the conclusions are interesting but require a minor revision before being published.

Author Response

Response to Reviewers’ Comments

Dear Editors and Reviewers:

Thank you for your letter and for the reviewers’ comments concerning our manuscript entitled “Enhanced Laser-induced Breakdown Spectroscopy for Heavy Mental Detection in Agriculture: A Review” (ID: sensors-1818490). Those comments are all valuable and very helpful for revising and improving our paper, as well as the important guiding significance to our researches. We have studied comments carefully and have made correction which we hope meet with approval. The revised part is marked out in the manuscript using in red. We answer every comment of reviewers after carefully studying the advices.

We appreciate for Editors/Reviewers’ warm work earnestly, and hope that the correction will meet with approval for publication. Once again, thank you very much for your comments and suggestions.

If you need any other information, please contact me immediately by e-mail. Our e-mails are [email protected] and [email protected].

Response to Reviewer 1:

Reviewer #1: In this work, authors illustrated the problems of LIBS technology in heavy metal detection in agriculture, introduced the methods of enhanced LIBS technology in terms of principles, and summarized the applications of the enhanced LIBS in heavy metal detection in agriculture. In my opinion, the field of writing is pertinent to this journal, the classification of enhancement methods in the review is relatively novel, the methodological discussion is detailed, and this work has wide application potential. Therefore, I think this paper is suitable for publication with some small modifications. Here are some suggestions and comments:

  1. Note the details of the abbreviation, line 16 "limits of detection" should be changed to "LOD".

Answer: Thanks for your valuable comment. It has been changed.

  1. Note the correctness of the tense. In line 21, "has" should be changed to "will have".

Answer: Thanks for your comment. It has been changed.

  1. In line 38, you should explain in detail whether the loss of characteristic peaks is characteristic of trace elements or whether the characteristic peaks of trace elements are more affected.

Answer: Thanks for your suggestion. They have been added in the appropriate positions.

  1. The "and" in line 108 should be deleted.

Answer: Thanks for your significant suggestion. It has been revised in the text.

  1. In line 121 you need to explain in detail what kind of sample parameters to find.

Answer: Thanks for your suggestion. It has been added in the revised version.

  1. In line 149, the stability of the laser-sample contact is increased rather than the area.

Answer: Thanks for your pertinent suggestion. It has been revised in the text.

  1. In line 208, it should be further clarified that orthogonal double-pulse have better detection performance than what.

Answer: Thanks for your comment. It has been modified in the text and added to the appropriate location.

  1. All tables in the review show the enhanced LIBS method and what the results are, but are not specific enough, so hopefully it can be added whether portable measurements are possible.

Answer: Thanks for your guiding comment. Most of LIBS measurements today are done in the laboratory, and portable LIBS are under development. Therefore, the examples in this review were also done in the laboratory.

  1. Note the uniformity of wording in the review. In line 224, "-" should be added between "single pulse" and in line 226, "-" should be added between "double pulse".

Answer: Thanks for your valuable comment. It has been added in the review.

  1. While the paper is well structured, there are several sentences difficult to understand because of improperly used English. The authors need to have this paper edited by someone proficient in technical English.

Answer: Thanks for your comment, it has been changed.

In all, the work is well done, and the conclusions are interesting but require a minor revision before being published.

The manuscript has been resubmitted to your journal. We look forward to your positive response.

Sincerely,

< Zihan Yang, Jie Ren, Mengyun Du, Yanru Zhao, Keqiang Yu >

Reviewer 2 Report

LIBS is one of the most complex spectroscopy technologies, but also has a huge potential for in-vivo agriculture applications. Most LIBS researchers focus their attention on hardware and method solutions for signal enhancement, and this is well documented in the review. The review is very important from this point of view, which may also be the area of expertise of the authors.

The manuscript is very well written and organized. I found no dificulty in the reading and was perfectly clarified at first reading. I find this review extremely valuable and important and should be published ASAP as is.

Author Response

Response to Reviewers’ Comments

Dear Editors and Reviewers:

Thank you for your letter and for the reviewers’ comments concerning our manuscript entitled “Enhanced Laser-induced Breakdown Spectroscopy for Heavy Mental Detection in Agriculture: A Review” (ID: sensors-1818490). Those comments are all valuable and very helpful for revising and improving our paper, as well as the important guiding significance to our researches. We have studied comments carefully and have made correction which we hope meet with approval. The revised part is marked out in the manuscript using in red. We answer every comment of reviewers after carefully studying the advices.

We appreciate for Editors/Reviewers’ warm work earnestly, and hope that the correction will meet with approval for publication. Once again, thank you very much for your comments and suggestions.

If you need any other information, please contact me immediately by e-mail. Our e-mails are [email protected] and [email protected].

Response to Reviewer 2:

Reviewer #2: LIBS is one of the most complex spectroscopy technologies, but also has a huge potential for in-vivo agriculture applications. Most LIBS researchers focus their attention on hardware and method solutions for signal enhancement, and this is well documented in the review. The review is very important from this point of view, which may also be the area of expertise of the authors.

The manuscript is very well written and organized. I found no difficulty in the reading and was perfectly clarified at first reading. I find this review extremely valuable and important and should be published ASAP as is.

Answer: Thank you for your affirmation. Based on the preliminary study, we will further study it in our future work.

Special thanks for your valuable comments.

The manuscript has been resubmitted to your journal. We look forward to your positive response.

Sincerely,

< Zihan Yang, Jie Ren, Mengyun Du, Yanru Zhao, Keqiang Yu >

Reviewer 3 Report

1. The language should be polished.

2. The range of this review in time is not clear, it is better have a range of time. For example, reference no. 60 is from 2012, and reference No. 61 is from 2013, reference No. 62 is from 2010. Although these articles may be interesting, they are a little old, so the review should has a certain timeliness. It is suggested to update the literature.

3. In part 2, there are many machine learning methods used to calibrate and analyze the LIBS data such as PCA, PLS, SVM and etc. It is necessary to introduce the necessity and advantages and limitations of these statistical techniques.

4. Let's unify the units. For example, in table 1, ppm and mg/L is equal, so it is necessary to unify the units in all tables and the whole text.

5. In table 2, reference 75, R2=00.99326? Authors should check all this paper to avoid similarly mistakes.

6. There are many repetitious sentences in the text. For example, in abstract,LIBS technique has been widely used for rapid detection of heavy metals with its advantages of convenient operation, simultaneous detection of multi-elements, wide range of elements, and no requirement for the state and quantity of samples.and in introduction, LIBS technique has attracted much attention because of its convenient operation, simultaneous detection of multi-elements, wide range of elements, no requirement for the state and quantity of samples, and so on.

“Therefore, the key to enhancing the LIBS signal is how to excite more plasma and dissipate less plasma both.” and “Only by exciting more plasma or scattering less plasma can the spectrometer pick up a stronger signal.” are mean the same meaning. Introduction section should be reorganized in the language.

7. There is no comparative analysis of traditional detection methods for heavy metals in agricultural products. 

8. In table 2, The number of decimal places is not uniform. 0.9931, 0.99882 and 0.991 etc. Table 1 and table 3 have the same error.

9. Page 6,line 222. Compared with the conventional method, the spectral intensity of Cd and Pb was significantly enhanced Lazaro, et al. [67] used dry ashing to prepare samples of plant leaves, and then carried out LIBS detection and analysis. Is it one sentence?

10. The first abbreviations should be spelled out in full.Example, page 8,line 288, LIBS-LIF, etc, please check all this in text.

11. In 5.3 part, the example is less, suggesting research more examples to illustrate this method

12. In page 13, line 433-434, it is highly recommended that this paragraph should be merged with other paragraphs.

13. The “Conclusion and future prospect” section is shallow, it should be revised because it is not deep enough.

14. Typographical errors-I spot some typographical errors, please read through the whole manuscript before submitting the revised version. Examples, in reference 2, 6,13,14, 10, 16, 17, 20, 21, 22, 23, 24, 25, 28, 30, 31, 35, 36,37, 38,40, 42,44, etc. All these reference are lack of page numbers. In reference 5, it should (2019), 223, 117374. 

15. LASER TECHNOLOGY , Spectroscopy and Spectral Analysis Appl. Phys. Lett.. Why there are three forms in your reference? The format of literature citation is inconsistent. So the whole paper is just like a first draft, which needs extensive revision. Please uniform the format according to the guidelines.

Author Response

Response to Reviewers’ Comments

Dear Editors and Reviewers:

Thank you for your letter and for the reviewers’ comments concerning our manuscript entitled “Enhanced Laser-induced Breakdown Spectroscopy for Heavy Mental Detection in Agriculture: A Review” (ID: sensors-1818490). Those comments are all valuable and very helpful for revising and improving our paper, as well as the important guiding significance to our researches. We have studied comments carefully and have made correction which we hope meet with approval. The revised part is marked out in the manuscript using in red. We answer every comment of reviewers after carefully studying the advices.

We appreciate for Editors/Reviewers’ warm work earnestly, and hope that the correction will meet with approval for publication. Once again, thank you very much for your comments and suggestions.

If you need any other information, please contact me immediately by e-mail. Our e-mails are [email protected] and [email protected].

Response to Reviewer 3:

Reviewer #3:

  1. The language should be polished.

Answer: Thanks for your valuable comment. It has been polished.

  1. The range of this review in time is not clear, it is better have a range of time. For example, reference no. 60 is from 2012, and reference No. 61 is from 2013, reference No. 62 is from 2010. Although these articles may be interesting, they are a little old, so the review should have a certain timeliness. It is suggested to update the literature.

Answer: Thanks for your pertinent comment. This review summarizes studies from the last decade or so. With regard to the references marked 60, 61, and 62, there are particularly few studies in this area, so they are also of reference interest.

  1. In part 2, there are many machine learning methods used to calibrate and analyze the LIBS data such as PCA, PLS, SVM and etc. It is necessary to introduce the necessity and advantages and limitations of these statistical techniques.

Answer: Thanks for your suggestion. They have been added in the appropriate positions.

  1. Let's unify the units. For example, in table 1, ppm and mg/L is equal, so it is necessary to unify the units in all tables and the whole text.

Answer: Thanks for your guiding suggestion. All units have been unified.

  1. In table 2, reference 75, R2=00.99326? Authors should check all this paper to avoid similarly mistakes.

Answer: Thanks for your comment. It has been changed.

  1. There are many repetitious sentences in the text. For example, in abstract, “LIBS technique has been widely used for rapid detection of heavy metals with its advantages of convenient operation, simultaneous detection of multi-elements, wide range of elements, and no requirement for the state and quantity of samples.” and in introduction, “LIBS technique has attracted much attention because of its convenient operation, simultaneous detection of multi-elements, wide range of elements, no requirement for the state and quantity of samples, and so on”.

“Therefore, the key to enhancing the LIBS signal is how to excite more plasma and dissipate less plasma both.” and “Only by exciting more plasma or scattering less plasma can the spectrometer pick up a stronger signal.” are mean the same meaning. Introduction section should be reorganized in the language.

Answer: Thanks for your suggestion. It has been revised in the text.

  1. There is no comparative analysis of traditional detection methods for heavy metals in agricultural products.

Answer: Thanks for your sincere suggestion. It has been added in the revised version.

  1. In table 2, The number of decimal places is not uniform. 0.9931, 0.99882 and 0.991 etc. Table 1 and table 3 have the same error.

Answer: Thanks for your comment. They have been unified.

  1. Page 6, line 222. “Compared with the conventional method, the spectral intensity of Cd and Pb was significantly enhanced Lazaro, et al. [67] used dry ashing to prepare samples of plant leaves, and then carried out LIBS detection and analysis.” Is it one sentence?

Answer: Thanks for your pertinent suggestion. It has been revised in the text.

  1. The first abbreviations should be spelled out in full. Example, page 8, line 288, LIBS-LIF, etc, please check all this in text.

Answer: Thanks for your comment. It has been spelled out in full on page 6, line 162.

  1. In 4.3 part, the example is less, suggesting research more examples to illustrate this method.

Answer: Thanks for your suggestion. However, there are only several studies about MA-LIBS in total, and only one of them is eligible for this review.

  1. In page 13, line 333-334, it is highly recommended that this paragraph should be merged with other paragraphs.

Answer: Thanks for your suggestion. It has been changed.

  1. The “Conclusion and future prospect” section is shallow, it should be revised because it is not deep enough.

Answer: Thanks for your valuable comment. It has been rewritten some details in this part.

  1. Typographical errors-I spot some typographical errors, please read through the whole manuscript before submitting the revised version. Examples, in reference 2, 6,13,14, 10, 16, 17, 20, 21, 22, 23, 24, 25, 28, 30, 31, 35, 36,37, 38,40, 42,44, etc. All these references are lack of page numbers. In reference 5, it should (2019), 223, 117374.

Answer: Thanks for your comment. It has been revised in the text.

  1. “LASER TECHNOLOGY”, “Spectroscopy and Spectral Analysis” “Appl. Phys. Lett.”. Why there are three forms in your reference? The format of literature citation is inconsistent.

Answer: Thanks for pertinent your comment. It has been revised in the text.

So, the whole paper is just like a first draft, which needs extensive revision. Please uniform the format according to the guidelines.

The manuscript has been resubmitted to your journal. We look forward to your positive response.

Sincerely,

< Zihan Yang, Jie Ren, Mengyun Du, Yanru Zhao, Keqiang Yu >

Round 2

Reviewer 3 Report

The authors have greatly improved and revised the paper based on my review comments, and I would happily recommend it for publication.